# Sport Nutrition Knowledge, Attitudes, Sources of Information, and Dietary Habits of Sport-Team Athletes

**DOI:** 10.3390/nu14071345

**Published:** 2022-03-23

**Authors:** Karla Vázquez-Espino, Gil Rodas-Font, Andreu Farran-Codina

**Affiliations:** 1Department of Nutrition, Food Science, and Gastronomy, XIA–INSA, Faculty of Pharmacy, Campus de l’Alimentació de Torribera, University of Barcelona, Av. Prat de la Riba, 171, E-08921 Santa Coloma de Gramenet, Barcelona, Spain; karlavespino@gmail.com; 2FC Barcelona Medical Services, Avda. Onze de Setembre, s/n, E-08970 Sant Joan Despí, Barcelona, Spain; gil.rodas@fcbarcelona.cat

**Keywords:** sport nutrition knowledge, nutrition attitude and practices, sources of nutrition information

## Abstract

Nutrition knowledge (NK) is one of several factors needed to establish proper eating habits and is especially important for athletes. The aims of this study were the following: to assess the NK of athletes from the Fútbol Club Barcelona; and to study its possible association with self-perceived level of NK, attitude towards nutrition, sources of information, and some dietary habits. We performed a cross-sectional study in two parts. First, we assessed the NK of elite athletes (*n* = 264) and compared it to the NK of technical teams of different sports (*n* = 59) and non-athletes (*n* = 183) of different ages and levels of education. Second, we investigated the associations between NK and other variables. To assess NK, we used a previously validated questionnaire Nutrition Knowledge Questionnaire for Young and Adult Athletes (NUKYA). Athletes showed a low median score (25.1 points), similar to the scores obtained by high school students (19.5) and university Philosophy students (29.0), and significantly lower than the scores of the sports technical team (58.5, p<0.05) and final year students of Human Nutrition and Dietetics (74.6, p<0.05). Moreover, we found statistically significant associations between NK and self-perceived level of NK (n=240,ρ=0.2546,p=0.0001) intake of fruits and vegetables (n=111,ρ=0.2701,p=0.0041), and intake of discretionary food (n=111,ρ=−0.2008,p=0.0001). Athletes with lower scores tended to overestimate their competence in nutrition (Dunning-Kruger effect). We concluded that NK of athletes needs to be improved through education plans that should consider aspects such as the proper selection of information resources and the importance of not consuming supplements without the adequate prescription. Incorporation of technical team and families to the education plan should be considered.

## 1. Introduction

Nutrition is considered a vital factor to stay healthy and well-being, as well as to optimize athletic performance. Nutrition knowledge (NK) is relevant to any sports player, especially to elite sports players. It consolidates the rationales to improve dietary behavior and grant the necessary competencies to make nutrition-based food choices. There is a wide consensus about the positive effect of a high NK in increasing fruits and vegetables intake [1,2] and decreasing fat consumption [3].

The analysis of NK among athletes has been primarily carried out with questionnaires designed to assess general nutrition knowledge [1,4,5,6,7], or specific sport-nutrition knowledge [8,9,10,11,12,13], and surveys for individual sports [14,15]. Most studies that compared athletes across all levels of sport engagement to other populations showed that the NK scores of athletes were lower or not different from non-athletic populations [4,5,16,17,18,19]. Unfortunately, no firm conclusions can be drawn as the measuring tools vary between studies [20]. Previous studies positively correlated NK to attending health and nutrition courses [6,9,21]. Moreover, athletes seem to have a limited knowledge about energy density, micronutrients, supplementation, diet sources of fat, muscle physiology, protein supplementation for vegetarians, and weight-loss management [22]. Other researchers found a positive correlation between NK scores and higher carbohydrate intake [23] with a lower protein to carbohydrate ratio (not reaching suboptimal levels) [5]. Likewise, positive associations between NK, fat-free mass, and weight maintenance are reported for athletes and elite players [5]. The latest systematic review about NK among athletes and coaches reveals that the majority of studies did not find significant differences on the basis of sex or sport types [22].

Many factors motivate food selection and dietary behavior [24,25,26]. Common factors like cultural background, taste preferences, appetite, attitude towards nutrition, and NK apply to general and athletic populations [27]. Furthermore, changes in body composition or aesthetics can influence the dietary behavior of an athlete [28,29]. Finally, for adolescent athletes, peer pressure and teammates’ choices seem to be important influencing elements [30]. The ways dietary behavior can influence performance are understudied, because of two main reasons: challenging quantification of dietary intake in a significant manner during a proper period of time; and the need to adequately assess NK with validated questionnaires in samples that are big enough [20,22,31].

In addition to compulsory education as a main source of NK, athletes rely on other sources like media, parents, friends, peers, coaches, and strength and conditioning staff [11,13,17,32,33]. All these sources can influence the dietary behaviors of athletes, sometimes providing limited or inaccurate nutritional information [12,34,35,36]. Previous research demonstrated that the lack of knowledge and the dissemination of erroneous nutritional guidance is harmful, especially for athletes [37]. In addition, individual factors, like the attitude towards nutrition, are not fully understood, representing a fruitful field for investigation and intervention.

In this study, our aims were the following: to assess the NK of athletes from the Fútbol Club Barcelona (FCB); and to explore the differences in NK between different teams, the technical staff of the club, and a sample of non-athletes of different ages and levels of education. Also, we aimed to study the possible association of NK with self-perceived level of NK, attitude towards nutrition, sources of information, and some dietary habits in athletes. All this information will be useful to prepare nutrition education programs for athletes.

## 2. Material and Methods

This is a cross sectional descriptive study to assess actual and self-perceived NK, attitudes, sources of information, and some dietary habits, in a sample of athletes and non-athletes. The study protocol was approved by the Ethics Committee for Clinical Research of the Catalan Sports Council. After obtaining collaboration approval with the institutions (sports club, university, and high school), volunteers >13 y or their tutors, in case they were <18 y, received verbal and written information about the purpose and procedures of the study. They all signed an informed consent before starting the study. All surveys were taken during the 2016–2017 season.

### 2.1. Participants

We asked athletes of four elite/sub-elite sports sections of FCB (soccer, roller hockey, indoor soccer or futsal, and basketball) and the sport technical team, STT (coaches, physical trainers, physiotherapists, and physicians), to voluntarily participate. All the mentioned sports sections have professionals teams in the elite of European sports clubs competitions. Groups of people that were neither athletic nor formally-educated in nutrition included high school students (HSS) and philosophy students from the University of Barcelona (UB-PHIL). We selected these two groups because their ages were in the range of most athletes in the sample (13–25 years) and they had a different level of education but had not received any formal education in nutrition. Finally, a group of non-athletes with a formal nutritional education was formed by students attending the final year of Human Nutrition and Dietetics at the UB (UB-HND).

### 2.2. Study Development

We conducted the study in two phases:

Phase I: Assessment of knowledge in sport nutrition. We assessed NK by using a previously validated short sport Nutrition Knowledge Questionnaire for Young and Adult Athletes (NUKYA) [31]. The NUKYA questionnaire consists of 20 questions that include a total of 59 items that assess the NK in four main domains: macronutrients, micronutrients, hydration, and periodization (food intake and hydration before and during exercise). The maximum possible scores for each section were the following: macronutrients, 49.1 points; micronutrients, 32.2; hydration 13.6; and periodization, 5.1 (see Appendix A). The questionnaire was self-administered and filled online using forms developed with Open Data Kit on Kobo Toolbox platform, and under the supervision of the main researcher to avoid discussion and interchange of information between respondents.

At the beginning of the survey, participants were requested to rank their perceived level of nutritional knowledge. Answers were provided in a Likert scale with five grades: 1, poor; 2, below average; 3, good on average; 4, very good; and 5, excellent. Then, participants were asked to complete the NUKYA questionnaire indicating some additional details (sex, date and place of birth, education level, and nationality).

Phase II: Assessment of attitude towards nutrition, sources of information and dietary practices in athletes. We had limited access to FCB athletes, not to alter their training schedule or interfere with their educational activities. Therefore, a subsample of 111 athletes was selected using stratified randomization, and we asked them to voluntarily answer a survey on the attitude towards nutrition, sources of information, and dietary practices. The survey was created according to the guidelines of the Knowledge-Attitudes-Practices (KAP) manual to assess nutrition-related knowledge, attitudes, and practices [38].

(a)Attitude towards nutrition. Sports players were asked if they thought that modification of their dietary habits could enhance their performance. The question was answered using a 5-point Likert scale format (1, definitely no; 2, probably no; 3, perhaps; 4, probably yes; and 5, definitely yes).(b)Sources of information. Participants were requested to select their sources of nutritional information from a non-rated list of ten options: social media (Instagram, Pinterest, websites, Facebook); magazines; friends; family; coaches; physiotherapists; books; scientific journals; dietitians; or others. In addition, they were asked if they had actively searched for nutritional advice from a professional source, and if yes, which one.(c)Dietary practices. Food consumption was evaluated through a semi-quantitative non-validated food frequency questionnaire (FFQ) based on a one-week intake. The FFQ included a list of 18 foods like fruit, pastry & biscuits, sweets & confectionery, and others. Liquid intake was evaluated with the BEVQ-15 [39], a brief validated quantitative questionnaire to assess habitual beverage intake. Food consumption frequencies were finally categorized as follows: never consumed, once a week, 2–3 times per week, 4–6 times per week, 1–2 times per day and more than 2 times per day. Frequencies of fruit and vegetable intake were summed to obtain an aggregated value. Also, intake frequency of discretionary foods was calculated by adding up the values of pastry & biscuits, sweets & confectionery, and sodas & energetic beverages. We also asked about the place where the three main meals and morning/afternoon snacks were consumed with a 4-option checkbox list: (1) I do not consume this meal, (2) out of home, (3) at the club facilities, (4) at home. Time of consumption was only recorded for the 3 mean meals (breakfast, lunch, and dinner). Finally, supplement intake was assessed in 2 steps: first, subjects were asked for the consumption of supplements using a binary question (yes/no); if the answer was positive, we asked the type and frequency of consumption.

### 2.3. Statistical Analysis

The analysis of data was performed using the STATA statistical software (version 16.1, StataCorp, College Station, TX, USA). The responses from the NUKYA questionnaire were directly recorded into a database, then scored as follows: a score of 0 was applied to “not sure/do not know” options, a positive score of 1 to correct answers, and a negative score of −1 to wrong answers. Total score and sub-scores were calculated by re-scaling the total score to a 0–100 range. First, we screened the data for missing values. Then, data normality was assessed with the Shapiro-Wilk test and homoscedasticity with the Levene’s or the Bartlett’s tests for equal variances. The strength of association between variables was tested with Spearman correlations. Means were compared using Student’s *t*-test or alternatively, Mann-Whitney’s test. Frequencies were compared using the chi-square test or the Fisher’s exact test, Kruskal-Wallis tests were used to compare groups when appropriate, followed by Bonferroni’s or Dunn’s post hoc tests respectively. Some relationships between variables were studied using analysis of covariance. For all the analyses, the confidence level was set at 95%.

## 3. Results

### 3.1. Sample Characteristics

A total of 506 participants, 146 women and 360 men, from five different groups (FCB, STT, HSS, UB-PHIL, and UB-HND) were evaluated for their NK. Underage participants (<18 years old) represented the 35.5% of the sample, with a mean age of 15.3 ± 1.3 years, while the rest of participants (64.5%) had a mean age of 25.2 ± 6.9 years. The total number of athletes from the FCB was 264, while non-athletes groups comprised a total of 242 subjects. Details on sex and age distribution are shown in (Table 1). In relation to the level of education, 41.4% had reached primary school, 45.3% high school or vocational training, and 13.3% university (bachelor’s or master’s degree, but no in nutrition and dietetics). Most athletes lived with their families (39.6%) or in a shared flat (34.0%); 23.6% lived in the FCB facilities; and only 2.8% lived alone. The majority of the athletes were Spanish (84%).

In relation to the random subsample used in the Phase II of the study, 111 athletes agreed to answer the questions about attitude towards nutrition, sources of information, and dietary practices. When comparing this group with the rest of the athletes (*n* = 153), we did not observe any differences in the age, level of studies, and nationality. However, the two groups differed in sex (20% of women in the subsample versus 4.5% among the rest of the athletes, $\chi^{2}$[1] = 13.597, *p* < 0.001), and in the NUKYA score (z=−3.233,p=0.0012). Indeed, the median score in the selected subsample was only five points higher than in the rest of athletes (23.7 points). Regarding the place where to have meals, 77% of the athletes in the subsample had breakfast at the sports club facilities, 18% at home, and 5% elsewhere. Thirty nine percent did not have a snack at the morning, 32% had a snack at the sport club facilities, 16% at home and 14% elsewhere. Most of the athletes (78%) had lunch at the sports club facilities, 17% at home, 4% elsewhere and 1% skipped this meal. Forty two percent had the afternoon snack at the club facilities, 42% at home, 14%, elsewhere and 2% skipped this meal. Finally, 58% of athletes had dinner at home, 41% at the club facilities, and 1% elsewhere.

### 3.2. Knowledge in Sport Nutrition

Median and interquartile range of the NUKYA scores were calculated (Table 2). Some data did not follow a normal distribution; therefore, all groups were compared using the Kruskal-Wallis test and the Dunn’s pairwise test with Bonferroni adjustment. The groups presented statistical differences between the total scores (*p* = 0.0001) and the partial scores: (*p* = 0.0001), such as: micronutrients (*p* = 0.0001); hydration (*p* = 0.0001); and periodization (*p* = 0.0001). The scores of the STT and UB-HND groups were significantly higher than those of the others. Moreover, the UB-HND scores were significantly higher than those of the STT (Table 2). When analyzing the FCB subgroups (basketball, soccer, futsal, and hockey), significant differences were detected in total (*p* = 0.0200), macronutrient (*p* = 0.0103), and periodization (*p* = 0.0073) scores. Details can be found in Table 3.

Since sex was not equally distributed among teams, differences between men and women in total and partial NUKYA scores could not be evaluated. However, when comparing the mean score of the women’s (*n* = 30, 31.8±3.52 points) and men’s (*n* = 42, 26.4±2.42 points) soccer teams, whose members were approximately the same age (23.6 and 21.1 years old, respectively), we did not detect statistically significant differences.

Spearman correlation coefficients were calculated between the NUKYA score of athletes and five variables, such as: age, self-perceived level of NK, attitude towards nutrition, frequency of fruits and vegetables intake, and frequency of discretionary foods intake. We obtained significant coefficients for four out of five variables: age (*n* = 263, 1 missing value, ρ=0.1809, *p* = 0.0032); self-perceived level of NK (*n* = 240, 24 missing values, ρ=0.2546, CI 95% 0.1263–0.3829, *p* = 0.0001); fruits and vegetables intake (*n* = 111, ρ=0.2701, *p* = 0.0041); and discretionary foods intake (*n* = 111, ρ=−0.2008, *p* = 0.0001). No significant correlation was found between NUKYA score and attitude towards nutrition. The Spearman coefficient between NUKYA score and the self-perceived level of NK in non-athletes (*n* = 198, 44 missing values, ρ=0.2682, CI 95% 0.1262–0.4102, *p* = 0.0346) was similar to the one in athletes.

Athletes were classified into five NK categories (very low, low, medium, high, and very high), each one with a range of 20 points, according to NUKYA scores. Then, we compared the distribution of this classification with the distribution of the self-perceived level of NK (poor, below average, right on average, very good, and excellent). Most of the athletes (42.1%) classified themselves in the category of knowledge just above the one they belonged according to the established NUKYA score classification; 24.2% two categories higher; and 3.8% three (Table 4). Additionally, 25.4% of the athletes correctly classified themselves according to their level of knowledge, and only a 4.5% classified themselves in a lower category. Therefore, our data indicated that athletes tended to overestimate their NK. When the same analysis was applied to the STT (*n* = 59), the situation was the reverse: 50.7% of the subjects classified themselves at least one level lower than the one they belonged. Kruskal-Wallis and Dunn’s pairwise comparison indicated that self-perception of NK differed between people with different NUKYA scores: a lower level of NK was associated with a higher self-estimation of it. However, since NK and age are associated, it may be possible that age acted as a confounding factor in this relationship. So, an analysis of covariance was performed in athletes, introducing the relative categorization of self-perceived NK as dependent variable and the NUKYA score and age as independent variables. The model was statistically significant (*n* = 239, p<0.0001) and definitely indicated that there was an effect of NK on its relative categorization (p<0.0001) that is independent of the effect of age (p<0.0001). Finally, a lower level of NK is associated with a higher self-estimation of it, independently of age.

### 3.3. Attitudes, Sources of Information, and Dietary Practices

No statistically significant differences were detected in the NUKYA mean scores of athletes, either on the basis of their attitude towards nutrition, or by comparing those actively seeking nutritional advice (42%) to those who did not (*n* = 111). However, the group of athletes that went to the dietitian (13%) had a higher (*p* = 0.0451) NUKYA mean score (39.4 points), in comparison to those that did not (29.2 points). No differences were detected in the distribution of assistance to the dietitian by age or sport team. Finally, athletes that consumed supplements (36%) had higher (*p* = 0.0204) mean NUKYA scores (35.7 points) than those who did not (27.5 points). It could be that there was a higher frequency of supplement consumption among athletes attending the dietitian’s office, or that athletes with a more favorable attitude towards nutrition could consume more supplements. So, we investigated all possible relationships between supplements intake, attendance at the dietitian’s office, attitude towards nutrition, and active search for nutritional advise. Using Fisher’s exact test, three statistically significant positive associations were found between the following variables: supplements intake and attitude towards nutrition (*p* = 0.037); supplements intake and active search for nutritional advice (*p* = 0.018); and attendance at the dietitian’s office and active search for nutritional advice (*p* < 0.001). Concerning supplements, the most consumed were vitamins (70%) and minerals (37.5%). Other supplements were concentrates or extracts of foods (23%), proteins and amino acids (18%), omega 3 fatty acids (8%), creatine (8%), collagen/chondroitin (8%), and carnitine (3%).

Analysis of the influence of the education level on the NUKYA score in the subsample of 111 athletes indicated that there were differences between sport groups (*p* = 0.0011); then, pairwise comparison indicated that middle (*n* = 63, median = 22.3 points) and high-school students (*n* = 63, median = 21.5 points) had a statistically significant different NUKYA score (*p* < 0.00129) from participants with a master’s degree (*n* = 6, median = 53.4 points). NUKYA scores of athletes who have completed middle- and high-school were really closed to those of the HSS group, and values of athletes with master’s degree were close to the STT (Table 2). None of these athletes underwent regulated studies in nutrition and dietetics.

The sources of information mentioned by athletes as usually consulted included family (57%), dietitians (57%), physiotherapists (53%), coaches (49%), Internet (38%), friends (21%), magazines (10%), scientific journals (8%), and books (6%). We tested if there were differences in the sources of information declared by the athletes according to the age, sex, the level of education, NUKYA score quartile, and attendance at a dietitian’s consultation. We observed that athletes with a higher NUKYA score tended to consult more scientific journals (*p* = 0.010); athletes with a higher level of education used to consult more books and scientific journals (*p* = 0.001); and young athletes sought for advice from family and physiotherapists more than older athletes (*p* = 0.0002 and *p* = 0.0034 respectively). Surprisingly, we did not detect differences in NUKYA scores between athletes who reported consulting the dietitian as a source of information and those who did not. The same result was obtained with the rest of sources of information, except for Internet (*p* = 0.0022); books (*p* = 0.0013); and scientific journals (*p* < 0.0001). We did not detect differences in age and level of education between Internet users and non-users. The users of books and scientific journals were mostly athletes with a bachelors’ or master’s degree.

Finally, the athletes from the subsample (*n* = 111) were ranked in the four quartiles of the NUKYA score obtained from the overall sample (*n* = 264). As expected in a random sample, the number of athletes per quartile did not differ from a homogeneous distribution according to the chi-squared test. We compared the medians of the frequency of intake between quartiles to see if there were differences in the consumption of fruit and vegetables, and discretionary food (Table 5). Since the role of both types of food in a healthy diet was directly or indirectly addressed in the NUKYA questionnaire, we expected an association between the NUKYA score and the frequency of intake of these foods. In the case of fruits and vegetables, there were no differences between the Q1, Q2, and Q3 quartiles of NUKYA score, but between these three quartiles and Q4. For discretionary foods intake, only a difference between quartiles Q1 and Q4 was detected (Table 5).

## 4. Discussion

The main purpose of this study was to benchmark the sport NK of elite and sub-elite FCB athletes from four sport teams. Furthermore, we compared it to the knowledge of non-athletes groups. No differences were observed between athlete’s and those non-athletes’ groups among people of comparable ages and levels of education: athletes had a global NK score that lied between the one of HSS students and the one of UB-PHIL students. It is difficult to compare these results with those from other authors because of the variability in the composition of groups. Indeed, in some studies, the nutritional knowledge is higher in athletes than in non-athletes, and the opposite occurs in others [20].

Athletes’ scores were clearly inferior to the scores obtained by STT and nutritionists students from the UB-HND. People in the STT group are older and have a higher level of education than athletes, and this can explain their higher scores. These results coincide with previous observations from two studies that assessed the NK of athletes and coaches, where coaches scored better than athletes [11,17]. However, differences are bigger in our study, and this can be partially explained by the fact that STT group not only included coaches and physical trainers, but also health professionals like physiotherapists and physicians.The study of Nor Azizam et al. [40] used this same questionnaire in athletes with sport science background in a Malaysian university, who obtained a much higher average score (58.6%, SD = 10.2%). This average score was close to those obtained by the STT group in our study. However, most of the Malaysian athletes (67 out of 70) had attended a nutrition course. Additionally, they were aged >18 years and were between the first and third year of their university studies, while our athletes hadn’t taken any nutrition course, 66% were aged <18 years and very few (13.3%) had university degrees or were studying them.

According to our results and in agreement with other studies, older age is associated with higher NK [3,41,42], so it can act as a potential confounder when comparing groups. Indeed, older people had more opportunities to acquire knowledge and greater cognitive maturity, incorporating more complex concepts. Other studies had not found a significant association between NK and age [32], probably because the age range was narrower than in our sample. As in most studies [41,43,44,45,46,47,48], we did not detect differences between sexes, although the comparison could not be done for all the teams because the only sample of female athletes was the one of the soccer team.

The scores obtained by athletes in the macronutrients and micronutrients sections did not reach the 30% of the maximum. Comparatively, the scores of athletes were lower in the macronutrients and micronutrients sections than in the hydration and periodization sections, whose scores were above the overall mean. In the case of hydration, our observations coincide with some studies [34,40,49], but differ from others [9,11]. Athletes should clearly improve their NK, mainly in the areas of macronutrients and micronutrients, since it was similar to that of groups representing the community and far below the knowledge demonstrated by STT. This need for improvement is less pronounced in athletes of the hockey team, which had highest NK median score due to a much higher score in the macronutrients section of the questionnaire. Moreover, according to the available data, age and education do not seem to explain these differences. Contrary to our observations, four studies revised by Trakman et al. [22] reported no significant differences in the NK scores between athletes from different sports [6,13,44,47].

When studying the relationship between the NUKYA score and the self-perceived level of NK, adjusting by age and sex, some interesting observations arose. Athletes with lower NUKYA scores tended to perceive their NK level as higher than it really was. The overestimation of their own NK could induce individuals to incorporate incorrect dietary practices with the conviction that they are beneficial, without considering the need for advice from health/nutrition professionals. This could lead to these athletes having too much confidence in their ability and therefore overestimating their competence to decide some aspects of their diet; that is, the so-called Dunning-Kruger effect [50]. So, it is important to detect athletes with the lowest NK and, among these, athletes that overestimate their own NK. To our knowledge, the scientific literature published to date does not describe similar observations on the nutritional competence of athletes.

Regarding the relation between NK and dietary intakes, we specifically studied the intake of two groups of foods: fruits and vegetables; and discretionary foods. The consumption of fruits and vegetables is positively associated with healthy diets, whereas the consumption of discretionary foods is inversely associated with healthy diets, and the same associations were expected with NK. Analysis of Spearman correlation coefficients indicated a weak but significant association of NK both with fruits and vegetables intake (ρ=0.270), and with discretionary foods intake (ρ=−0.201). Our results are in agreement with previous scientific literature: r=0.36 for fruits and r=0.23 for vegetables [3], ρ=0.28 for fruits and vegetables, ρ=−0.29 for junk food [41], and ρ=0.225 for vegetables [51]. When we analyzed intakes aggregating data in quartiles, we observed that associations were confirmed in the extremes (Q1 and Q4), but the quartiles in the middle (Q2 and Q3) showed a plateau. The reason for this is difficult to glimpse with the available data.

Association between NK and healthy dietary practices has previously been reported either as weak or lacking. This could be due to limitations in the quality of the studies, and the adequacy and validity of the tools used [18]. However, the NUKYA questionnaire has been previously validated [31]. FFQ is described in scientific literature as a dietary assessment tool that correctly classifies subjects according to their level of consumption, although it is inevitably imprecise in assessing intake [18]. We used a FFQ that was not previously validated and this could affect the reliability of our results. In spite of all this, we cannot rule out that the association between NK and intake of fruits and vegetables or discretionary foods simply follows a plateau dynamic, which seems to be suggested by our data, and that could actually explain the weakness of the associations between these variables found in our study. Spronk et al. [18] observed that these weak associations could be explained by the lack of connectedness between the aspects inquired in the NK questionnaire and the assessment of dietary intake could explain the observed lack of association. In our case, this problem was avoided with a previous definition of the concepts that should be evaluated with the NUKYA questionnaire: we selected basic and practical concepts that had a direct link with decisions related to a healthy diet. In addition, items included in the questionnaire underwent a content validation by a panel of experts [31]. Finally, studies that use large samples and validated tools did detect significant associations between some aspect of the diet and NK, and this would encourage further research [18].

The level of education was positively associated with NK, as previously described [6,7,9,52,53,54]. Moreover, when athletes and non-athletes were aggregated according to their level of education, and subgroups with similar education levels compared for their NUKYA scores, such scores were coherent, supporting the association of education with NK. The main sources of information reported by athletes were families, dietitians, physiotherapists, coaches, and Internet; in general, these coincide with the sources reported in more recent studies [32,40,42] and are quite different from those reported in older studies [41,55]. Dietitians were reported as one of the main sources of information (57%), although only 13% of athletes consistently attended the dietitian office. Devlin and Belski [32], discussing a similar incoherence, mentioned the possibility that athletes would feel compelled to select “dietitian” as a source of information, although they did not necessarily seek advice from a professional. In our study, family represented an important source of information, probably because our sample had an important percentage of adolescents (30% < 15 y). So, the nutritional education of families (in the case of young athletes), physiotherapists, and coaches should be supported to avoid possible mistakes and inconsistencies in the information that athletes receive. In addition, Internet has become an important source of information. However, this information should be properly selected according to the solvency of the organization that publishes it; so, the education of athletes should include how to identify solvent Internet resources.

Attitude towards nutrition was assessed by asking participants if they thought that modifications in their dietary habits could enhance their performance, and it was not associated with the NUKYA score. No association was found neither with active search for nutritional advice. However, athletes that used to attend to the dietitian and those that consume supplements had higher NUKYA scores, but we did not detect any association between the two variables: the usual visit to the dietitian did not seem to increase the chances to consume supplements. Therefore, athletes seem motivated to consume supplements by the interest in improving their performance, rather than by the advice of a dietitian. This trend needs to be corrected because the intake of supplements should be prescribed and monitored by healthcare professionals.

This study has several limitations, some of them already mentioned. First, the sample was restricted to athletes from one sports club; however, it is one of the largest sports club in Europe, with five professional and nine amateur sports sections, three of which have women’s teams (one professional), with athletes from different regions and countries. Second, some tools used to capture information about attitudes or dietary practices were simple and/or not validated; however, more sophisticated tools require more time or are difficult to understand for younger subjects, and these aspects had to be considered in our case. Moreover, sometimes, there are no validated short questionnaires for the population of interest, as in the case of FFQ. Third, the phase II of the study could not be performed with the whole sample but with a random selection of athletes; however, this subsample has been demonstrated to be representative of the whole sample. Finally, another limitation was the small number of female athletes in the sample that did not allow to properly compare NK by sex, although this group did not showed differences from their male counterpart.

The main strengths of our study is that the sample size was high, and NK was also assessed in non-athletes of different ages and levels of education. This allowed to make more comparisons and extract more detailed conclusions about the athletes’ level of NK. Finally, we used our own validated NK questionnaire that has good psychometric properties, and this might be the explanation for the high consistency of the NK scores obtained.

## 5. Conclusions

According to the results of the NUKYA questionnaire, athletes need to improve their NK, especially in the areas of macronutrients and micronutrients, to be autonomous in their decisions and achieve a healthy diet. Additionally, our study provides some evidence that seems to indicate an association between NK and healthy dietary habits.

A nutrition education plan for athletes should be developed and it should incorporate, in addition to practical concepts for daily practice, aspects like the proper selection of information resources (especially from Internet). The plan should also be addressed to families in the case of younger athletes, and to the technical team. Dietitians or nutritionists should play a leading role as a source of information and advice for athletes and members of the sports technical team. Moreover, nutrition education should include information on supplements and emphasize the importance of not consuming them without a proper prescription. Also, we need to be vigilant and give priority to the case of athletes with low NK and an excessive level of confidence in it. These cases typically involve young individuals. Finally, although the study was conducted within a specific sports club, thanks to its size and characteristics, we think our findings could be extended to others.

## Figures and Tables

**Table 1 nutrients-14-01345-t001:** Sex and age distribution in athletes (n=264) and non-athletes’ (n=242) groups. Abbreviations: FCB, athletes of Fútbol Club Barcelona; HSS, high-school students; UB-PHIL, philosophy students; STT, sports technical team; UB-HND, Human and Nutrition Dietetics students.

Athletes from FCB		Basketball	Hockey	Soccer	Futsal
Number of subjects		65 (24.6%)	18 (6.8%)	166 (62.9%)	15 (5.7%)
Sex	Male	65 (100.0%)	18 (100.0%)	136 (81.9%)	15 (100.0%)
	Female	–	–	30 (18.1%)	–
Age rank (years)	Max–Min	19.6–37.8	15.5–23.8	13–29.9	17.3–36.3
Distribution by age	13–15	39 (60.0%)	5 (27.8%)	36 (21.7%)	–
	16–18	21 (32.3%)	6 (33.3%)	64 (38.6%)	3 (20.0%)
	19–21	3 (4.6%)	6 (33.3%)	34 (20.5%)	1 (6.7%)
	22–25	–	1 (5.6%)	17 (10.2%)	2 (13.3%)
	>25	2 (3.1%)	–	15 (9.0%)	9 (60.0%)
**Non Athletes**		**STT**	**UB-HND**	**UB-PHIL**	**HSS**
Number of subjects		59 (24.4%)	51 (21.1%)	39 (16.1%)	93 (38.4%)
Sex	Male	42 (71.2%)	15 (29.4%)	27 (69.2%)	42 (45.2%)
	Female	17 (28.8%)	36 (70.6%)	12 (30.8%)	51 (54.8%)
Age rank (years)	Max–Min	25–55	21.8–36.1	18.7–39.6	13.8–17
Distribution by age	13–15	–	–	–	88 (94.6%)
	16–18	–	–	1 (2.6%)	5 (5.4%)
	19–21	–	2 (3.9%)	7 (17.9%)	–
	22–25	–	34 (66.7%)	11 (28.2%)	–
	>25	59 (100%)	15 (29.4%)	20 (51.3%)	–

**Table 2 nutrients-14-01345-t002:** Medians (interquartile range) of the NUKYA total and partial scores for the questionnaire subsections in all groups (N=506). Different superscript letters indicate statistically significant differences (*p* < 0.05). Abbreviations: FCB, athletes of Fútbol Club Barcelona; HSS, high-school students; UB-PHIL, philosophy students; STT, sports technical team; UB-HND, Human and Nutrition Dietetics students.

Group	*n*	Total Score	Macronutrients	Micronutrients	Hydration	Periodization
UB-HND	51	74.6 ${(13.6)\;}^{a}$	39.0 ${(8.5)\;}^{a}$	22.0 ${(6.8)\;}^{a}$	8.5 ${(8.5)\;}^{a}$	5.1 ${(0)\;}^{a}$
STT	59	58.5 ${(13.6)\;}^{b}$	28.8 ${(15.3)\;}^{b}$	16.9 ${(7.6)\;}^{b}$	7.6 ${(3.1)\;}^{b}$	5.1 ${(1.7)\;}^{b}$
UB-PHIL	39	29.0 ${(22.2)\;}^{c}$	10.2 ${(16.9)\;}^{c}$	12.7 ${(8.5)\;}^{c}$	4.5 ${(5.1)\;}^{\mathrm{cd}}$	2.8 ${(2.3)\;}^{\mathrm{cd}}$
FCB	264	25.1 ${(21.3)\;}^{c}$	8.5 ${(13.6)\;}^{\mathrm{cd}}$	8.5 ${(8.5)\;}^{d}$	5.4 ${(5.1)\;}^{c}$	3.0 ${(3.8)\;}^{c}$
HSS	93	19.5 ${(22.0)\;}^{c}$	6.8 ${(10.2)\;}^{d}$	9.3 ${(9.3)\;}^{\mathrm{cd}}$	3.4 ${(3.1)\;}^{d}$	1.3 ${(3.0)\;}^{d}$

**Table 3 nutrients-14-01345-t003:** Medians (interquartile range) of the NUKYA total and partial scores for the questionnaire subsections in FCB athletes (N=264). Different superscript letters indicate statistically significant differences (*p* < 0.05).

Sports Sections	*n*	Total Score	Macronutrients	Micronutrients	Hydration	Periodization
Hockey	18	38.6 (14.8) ^a^	18.6 (10.2) ^a^	11.0 (10.2) ^a^	6.2 (4.8) ^a^	3.0 (3.4) ^a^
Futsal	15	28.0 (25.0) ^ab^	8.5 (11.9) ^b^	9.3 (6.8) ^a^	6.8 (7.3) ^a^	3.4 (3.8) ^a^
Basketball	65	24.0 (22.3) ^b^	10.1 (11.9) ^ab^	8.5 (7.6) ^a^	5.1 (4.8) ^a^	1.7 (2.7) ^b^
Soccer	166	23.7 (19.2) ^b^	6.8 (13,6) ^b^	8.1 (7.6) ^a^	5.2 (5.1) ^a^	3.1 (3.4) ^a^

**Table 4 nutrients-14-01345-t004:** Distribution of the deviation of the athletes self-perceived level of NK in relation to the NUKYA score classification. The last column includes the median of the NUKYA score for each group of athletes in the row. Different superscript letters indicate statistically significant differences in median scores (*p* < 0.05).

Frequencies	NUKYA Score Classification					Median Scores (P5%–P95%)
	Very Low	Low	Medium	High	Very High	
Self-perceived NK with respect						
to the NUKYA score classification						
*One level lower*	0	3	6	1	1	44.9 ^a^ (25.7–80.8)
*At the corresponding level*	16	21	23	1	0	30.4 ^b^ (1.7–55.1)
*One level higher*	23	71	6	1	0	27.3 ^b^ (9.9–41.4)
*Two levels higher*	48	9	1	0	0	14.1 ^c^ (1.3–36.7)
*Three levels higher*	9	0	0	0	0	7.1 ^c^ (1.3–17.8)
Total (*n* = 240, 24 missing values)	96	104	36	3	1	24 (1.7–52.5)

**Table 5 nutrients-14-01345-t005:** Medians (interquartile ranges) of NUKYA scores and frequencies of consumption of fruit and vegetables and discretionary foods in a random subsample of athletes (*n* = 111). In each column, different superscript letters indicate statistically significant differences between median frequencies of intake (*p* < 0.05).

NUKYA Score Quartiles	N	NUKYA Score	Fruits & Vegetable (per Week)	Discretionary Foods (per Week)
Q1	26	10.8 (10.3) ^a^	6.2 (7.0) ^a^	7.0 (9.5) ^a^
Q2	31	23.6 (7.3) ^b^	9.5 (11.5) ^a^	5.0 (7.0) ^ab^
Q3	39	38.4 (10.2) ^c^	9.0 (11.5) ^a^	6.0 (5.5) ^ab^
Q4	15	57.6 (20) ^d^	21 (17) ^b^	3.5 (4.0) ^b^

## Data Availability

The data presented in this study are available on request from the corresponding author. The data are not publicly available due to ethical.

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
