# Peer review of "Sport Nutrition Knowledge, Attitudes, Sources of Information, and Dietary Habits of Sport-Team Athletes"

_nutrients, 2022, doi:10.3390/nu14071345_

Round 1

Reviewer 1 Report

Major concerns

  1. Similar studies on athletes' lack of nutritional knowledge have been conducted, and the authors should explore strategies to improve athletes' nutritional knowledge.
  2. The reliability and validity of the questionnaire should be presented.
  3. In statistical analysis, it is not reasonable to compare the nutrition knowledge of high school students with that of professional teams.
  4. Were authors actually examining the association with gender or did the questionnaire assess biological sex (i.e. was a binary option presented to respondents or were they able to self select from a variety of gender identities?)
  5. The organization of Tables 2, 3 and 4 is not sufficiently clear and the comparisons in the statistical analysis are not clearly explained.
  6. Study limitations could be added to the last paragraph of the discussion.
  7. Line 148, authors mentioned “Relationships between variables were studied using multiple linear regression.“ Where is the result of this analysis presented in the text ?

Reviewer 2 Report

Review / Sport nutrition knowledge, attitudes, sources of information, and dietary habits of sport-team athletes

First of all, congratulation for very good paper. I think it is generally written very well and provides some new information regarding sport nutrition. However, some part of the manuscript should improve.

INTRODUCTION

Very well written and easy to follow.

METHOD:

Participants:

Please provide number of participants in each category.

Why exactly high school students (HSS) and philosophy students from the University of Barcelona were chosen?

Study development:

On which online platform was survey conducted?

When you are mentioning KAP for the first time, please write the full name.

It would be good to provide some example of NK questions so the readers fully understand the process of data collection and further analysis.

RESULTS:

No objections!

DISCUSSION:

Please highlight all major findings in the first chapter of the discussion.

Are there any studies on athletes that used NUKYA questionnaire? If yes, provide some details and comparison with them.

Page 9, line 340-347 – This is classical example of Dunning-Kruger effect. I think it would be very suitable to use this psychological phenomenon for explaining these findings.

Page 10, line 382-389 – There is no need to repeating results (numbers) again. Please amend the text. 

CONCLUSION:

Please add some details what to investigate in the future researches on this topic.

Round 2

Reviewer 1 Report

There are some minor concerns that I hope the authors will take note of.

  1. Please note that the revised changes should be highlighted.
  2. Table 2-5, “Different superscript letters indicate statistically significant differences (P<0.05).” Please indicate what a, b, c, d means for. (ex, < .05, < .01 or < .001)
  3. Please note whether the table title should be placed at the top of the table and a short description appear below the table.
  4. Please note the consistency of upper and lower case of P values. Ex, line 175, table 4.

Author Response

Thank you for your additional comments. Please see the attachment.
